# Residue-by-residue analysis of cotranslational membrane protein integration in vivo

**Felix Nicolaus[1†], Ane Metola[1†], Daphne Mermans[1†], Amanda Liljenström[1], Ajda Krč[1,2], Salmo Mohammed Abdullahi[1], Matthew Zimmer[3], Thomas F Miller III[3], Gunnar von Heijne[1,4*]**

[1]Department of Biochemistry and Biophysics, Stockholm University, Stockholm, Sweden; [2]Faculty of Chemistry and Chemical Technology, University of Ljubljana, Ljubljana, Slovenia; [3]California Institute of Technology, Division of Chemistry and Chemical Engineering, Pasadena, United States; [4]Science for Life Laboratory Stockholm University, Solna, Sweden

**Abstract** We follow the cotranslational biosynthesis of three multispanning *Escherichia coli* inner membrane proteins in vivo using high-resolution force profile analysis. The force profiles show that the nascent chain is subjected to rapidly varying pulling forces during translation and reveal unexpected complexities in the membrane integration process. We find that an N-terminal cytoplasmic domain can fold in the ribosome exit tunnel before membrane integration starts, that charged residues and membrane-interacting segments such as re-entrant loops and surface helices flanking a transmembrane helix (TMH) can advance or delay membrane integration, and that point mutations in an upstream TMH can affect the pulling forces generated by downstream TMHs in a highly position-dependent manner, suggestive of residue-specific interactions between TMHs during the integration process. Our results support the 'sliding' model of translocon-mediated membrane protein integration, in which hydrophobic segments are continually exposed to the lipid bilayer during their passage through the SecYEG translocon.

**\*For correspondence:**
Gunnar.von.Heijne@dbb.su.se

[†]These authors contributed equally to this work

**Competing interests:** The authors declare that no competing interests exist.

## Introduction

Most integral membrane proteins are cotranslationally integrated into their target membrane with the help of translocons such as bacterial SecYEG and YidC, and the eukaryotic Sec61 and EMC complexes (*Rapoport et al., 2017*; *Chitwood et al., 2018*). While the energetics of translocon-mediated integration of a transmembrane α-helix (TMH) is reasonably well understood (*Hessa et al., 2007*), the actual integration process is not, other than in general terms. We have shown that force profile analysis (FPA) – a method in which a translational arrest peptide (AP) engineered into a target protein serves as a sensor to measure the force exerted on a nascent polypeptide chain during translation – can be used to follow the cotranslational folding of soluble proteins and the membrane integration of a model TMH (*Ismail et al., 2012*; *Ismail et al., 2015*; *Farías-Rico et al., 2018*). Here, we have applied FPA and coarse-grained molecular dynamics (CGMD) simulations to follow the cotranslational membrane integration of three multispanning *Escherichia coli* inner membrane proteins of increasing complexity (EmrE, GlpG, BtuC), providing the first residue-by-residue data on membrane protein integration in vivo.

## Results

### Force profile analysis

FPA takes advantage of the ability of APs to bind in the upper parts of the ribosome exit tunnel and thereby pause translation when their last codon is in the ribosomal A-site (*Ito and Chiba, 2013*). The duration of an AP-induced pause is reduced in proportion to pulling forces exerted on the nascent chain (*Goldman et al., 2015*; *Kemp et al., 2020*), that is, APs can act as force sensors and can be tuned by mutation to react to different force levels (*Cymer et al., 2015a*). In an FPA experiment, a series of constructs is made in which a force-generating sequence element (e.g., a TMH) is placed an increasing number of residues away from an AP (reflected in $N$, the number of residues from the start of the protein to the end of the AP), which in turn is followed by a C-terminal tail (*Figure 1a*). In constructs where the TMH engages in an interaction that generates a strong enough pulling force $F$ on the nascent chain at the point when the ribosome reaches the last codon of the AP, pausing will be prevented and mostly full-length protein will be produced during a short pulse with [$^{35}$S]-Met (*Figure 1b*, middle). In contrast, in constructs where little force is exerted on the AP, pausing will be

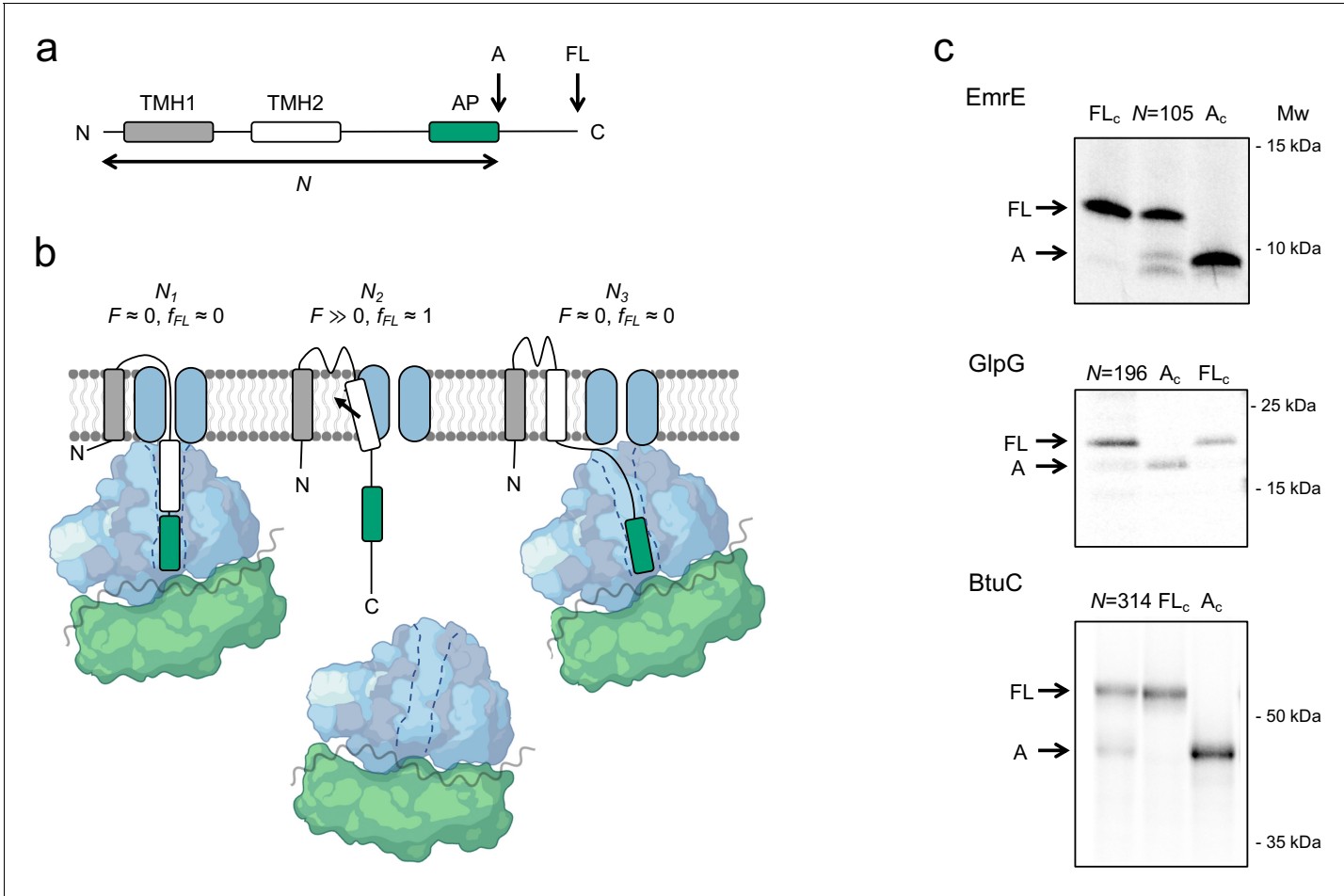

**Figure 1.** The force profile assay. (a) Basic construct. Arrested (*A*) and full-length (*FL*) products are indicated. (b) At construct length $N_1$, TMH2 has not yet entered the SecYEG channel and no pulling force $F$ is generated. At $N_2$, TMH2 is integrating into the membrane and $F \gg 0$. At $N_3$, TMH2 is already integrated and $F \approx 0$. (c) SDS-PAGE gels showing *A* and *FL* products for [$^{35}$S]-Met labeled and immunoprecipitated EmrE($C_{out}$) ($N$ = 105), GlpG ($N$ = 196), and BtuC ($N$ = 314). Control constructs $A_C$ and $FL_c$ have, respectively, a stop codon and an inactivating Ala codon replacing the last Pro codon in the arrest peptide (AP). The band just below the *A* band in the EmrE($C_{out}$) ($N$ = 105) lane most likely represents ribosomes stacked behind the AP-stalled ribosomes (*Notari et al., 2018*) and is not included in the calculation of $f_{FL}$. See *Figure 1—figure supplement 1* for additional gels. The online version of this article includes the following figure supplement(s) for figure 1:

**Figure supplement 1.** Gel gallery with selected EmrE (a–e), GlpG (f–l), and BtuC (m–r) constructs.

efficient and more of the arrested form of the protein will be produced (*Figure 1b*, left and right). The fraction full-length protein produced, $f_{FL} = I_{FL}/(I_{FL}+I_A)$, where $I_{FL}$ and $I_A$ are the intensities of the bands representing the full-length (*FL*) and arrested (*A*) species on an SDS-PAGE gel (*Figure 1c* and *Figure 1—figure supplement 1*), can therefore be used as a proxy for *F* in a given construct (*Kemp et al., 2020*; *Niesen et al., 2018*; *Leininger et al., 2019*). A plot of $f_{FL}$ versus *N* – a force profile (FP) – thus can provide a detailed picture of the cotranslational process in question, as reflected in the variation in the force exerted on the nascent chain during translation. FPs can be recorded with up to single-residue resolution by increasing *N* in steps of one residue (corresponding to a lengthening of the nascent chain by ~3 Å).

## EmrE: 4 TMHs, 110 residues

We chose EmrE as an example of a small, relatively simple 4-TMH protein. EmrE is a dual-topology protein, that is, the monomers integrate into the inner membrane in a 50–50 mixture of $N_{in}$-$C_{in}$ and $N_{out}$-$C_{out}$ topologies; two oppositely oriented monomers then assemble into an antiparallel dimer (*Chen et al., 2007*; *Rapp et al., 2007*). To avoid potential complications caused by the dual topology, we used EmrE($C_{out}$), a mutant version that adopts the $N_{out}$-$C_{out}$ topology (*Rapp et al., 2007*), and further used the relatively weak SecM(*Ec*) AP (*Ismail et al., 2012*) and included an HA tag for immunoprecipitation (*Figure 2a*). A series of EmrE($C_{out}$)-AP constructs (see *Supplementary file 1* for sequences) was used to obtain the FP shown in *Figure 2b* (orange curve), at 2–5 residues resolution. Also shown is an FP derived from a CGMD simulation (CGMD-FP, gray; *Van Lehn et al., 2015*); a hydrophobicity plot (HP) is included in *Figure 2—figure supplement 1*.

We have previously shown that a model TMH composed of Ala and Leu residues generates a peak in an FP recorded with the SecM(*Ec*) AP that reaches half-maximal amplitude ($N_{start}$) when the N-terminal end of the TMH is ~45 residues away from the polypeptide transferase center (PTC) (*Ismail et al., 2012*), and a recent real-time FRET study of cotranslational membrane integration found that the N-terminal end of the first TMH in a protein reaches the vicinity of the SecYEG translocon when it is 40–50 residues away from the PTC (*Mercier et al., 2020*). For EmrE($C_{out}$) TMH1, this would correspond to constructs with $N \approx 50$. However, the $f_{FL}$ values are hardly above background in this region of the FP. Due to the functionally important $E^{14}$ residue, TMH1 is only marginally hydrophobic and does not become firmly embedded in the membrane until the protein dimerizes (*Seurig et al., 2019*). To ascertain whether the lack of a peak in the FP corresponding to the membrane integration of TMH1 is because of its low hydrophobicity, we mutated $E^{14}$ to L. Indeed, in the FP obtained for EmrE($C_{out}$,$E^{14}$L) (*Figure 2b*, green curve), a clear peak appears at the expected chain length $N_{start} \approx 50$ residues. Mutation $E^{14}$A yields an $f_{FL}$ value intermediate between EmrE($C_{out}$,$E^{14}$L) and EmrE($C_{out}$) at $N = 55$ (*Figure 2c*), while $f_{FL}$ for the mutants EmrE($C_{out}$,$E^{14}$D) and EmrE($C_{out}$,$E^{14}$Q) is the same as for EmrE($C_{out}$).

Peak II has $N_{start} \approx 76$, corresponding to a situation where the N-terminal end of TMH2 is ~45 residues from the PTC (*Figure 2d*). The double mutation $I^{37}I^{38}{\rightarrow}$NN in TMH2 reduces $f_{FL}$ at $N = 80$ and 85 (magenta triangles), as expected. Unexpectedly, however, the $E^{14}$L, $E^{14}$A, and $E^{14}$Q (but not the $E^{14}$D) mutations in TMH1 increase $f_{FL}$ at $N = 85$ (*Figure 2c*), showing that a negatively charged residue (D or E) in position 14 in TMH1 specifically reduces the pulling force generated by TMH2 at $N = 85$, that is, when about one-half of TMH2 has integrated into the membrane. Likewise, $f_{FL}$ values at $N = 115$ and 130 (but not at $N = 105$, included as a negative control) are specifically affected by mutations in $E^{14}$: at $N = 115$ (one-half of TMH3 integrated), all four mutations in position 14 increase $f_{FL}$ relative to $E^{14}$, while at $N = 130$ (beginning of TMH4 integration) the $E^{14}$A and $E^{14}$L mutations decrease $f_{FL}$ (*Figure 2c*). FPA thus reveals long-range effects of mutations in $E^{14}$ on three specific steps in the membrane integration of the downstream TMHs. This implies that TMH1 remains in the vicinity of the translocon and that $E^{14}$ makes specific interactions with residues in the TMH2–TMH4 region during the membrane integration process. Further studies will be required to pinpoint these interactions and understand the role played by the slow dynamics of TMH1 integration (*Seurig et al., 2019*).

Peak III has $N_{start} \approx 102$ residues, with the N-terminal end of TMH3 ~45 residues from the PTC (*Figure 2d*). Peak IV is difficult to locate precisely in the FP because $f_{FL}$ values are high throughout the TMH3–TMH4 region, but is seen at $N_{start} \approx 132$ residues when the strong SecM(*Ec*-sup1) AP (*Yap and Bernstein, 2009*) is used (blue curve), again with the N-terminal end of TMH4 ~45 residues from the PTC (*Figure 2d*). As shown in *Figure 2e*, the TMHs cease generating a pulling force when

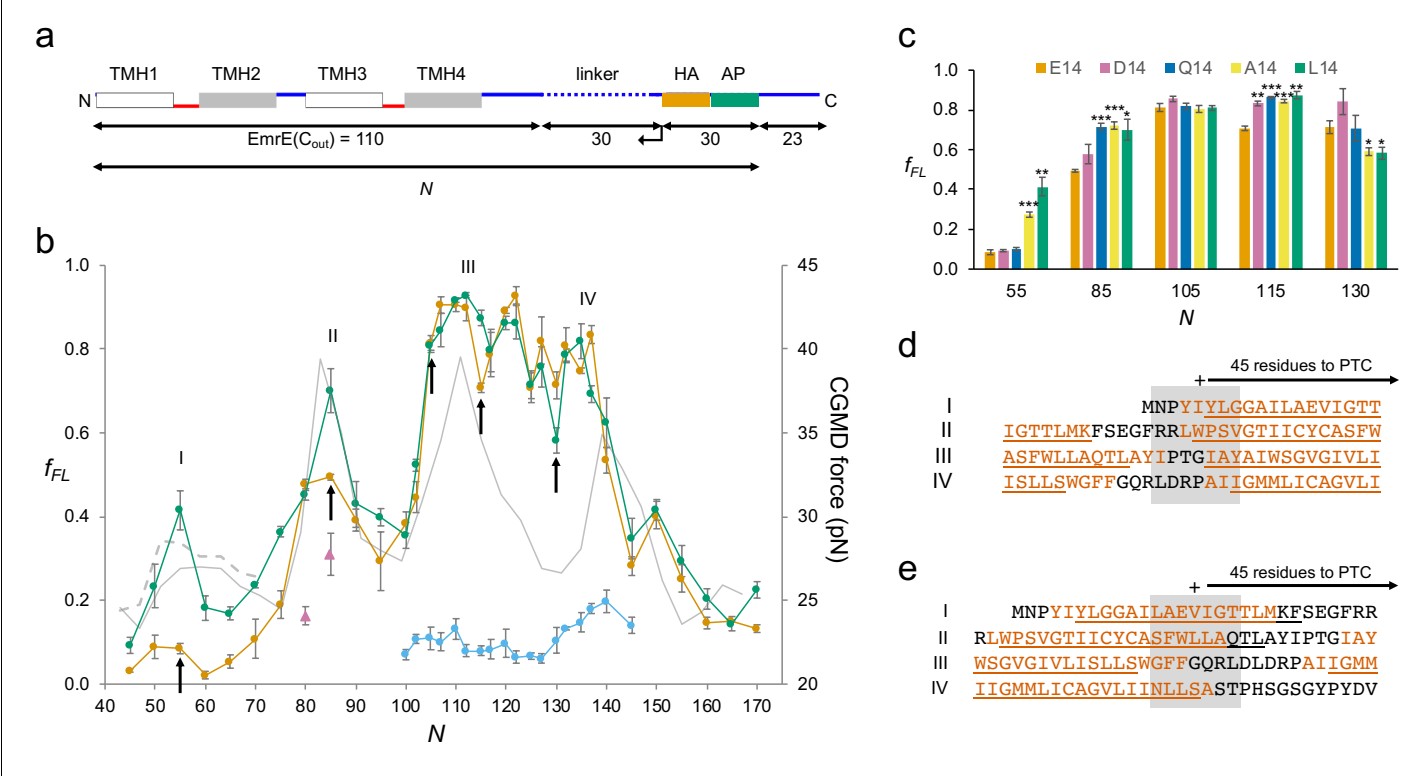

**Figure 2.** EmrE($C_{out}$). (a) Construct design. EmrE($C_{out}$) is shortened from the C-terminal end of the LepB-derived linker (dotted), as indicated by the arrow. Cytoplasmic (red) and periplasmic (blue) loops, and lengths of full-length EmrE($C_{out}$), LepB-derived linker, HA tag + arrest peptide (AP), and C-terminal tail, are indicated. Since the 30-residue HA + AP segment is constant in all constructs, the force profile (FP) reflects nascent chain interactions occurring mainly outside the ribosome exit tunnel. (b) FPs for EmrE($C_{out}$) (orange), EmrE($C_{out}$,$E^{14}$L) (green), EmrE($C_{out}$) with SecM($Ec$-sup1) AP (blue), EmrE($C_{out}$, $I^{37}I^{38}$→NN) (magenta triangles), and coarse-grained molecular dynamics (CGMD-FP) calculated with a −100 mV membrane potential (gray). (c) Effects of mutations in $E^{14}$ on $f_{FL}$ values for the $N$ values are indicated by arrows in (b). p-values (two-sided t-test): *p < 0.05; **p < 0.01; ***p < 0.001. (d, e) Sequences corresponding to peaks I–IV aligned from their $N_{start}$ (d) and $N_{end}$ (e) values. The + sign indicates 45 residues from the polypeptide transferase center (PTC). Hydrophobic transmembrane helix (TMH) segments are shown in orange and transmembrane α-helices underlined (PDB: 3B5D). Error bars in b and c indicate SEM values.

The online version of this article includes the following figure supplement(s) for figure 2:

**Figure supplement 1.** EmrE($C_{out}$).

their C-terminal ends are ~45 residues away from the PTC, indicating that they are fully integrated at this point.

## GlpG: 6 TMHs, 276 residues

We next studied GlpG, a medium-sized monomeric 6-TMH rhomboid protease with an ~60 residue cytoplasmic N-terminal domain (NTD) (*Sherratt et al., 2012*; *Wang et al., 2006*) (*Figure 3a)*, a protein that allows us to follow the cotranslational folding of a soluble domain and integration of a membrane domain in the same experiment.

The FP is shown in *Figure 3b* (orange curve). It was obtained at 5-residue resolution, except for the portion $N$ = 168–224, which we measured with single-residue resolution. For unknown reasons, constructs with $N$ ≈ 140–190 residues gave rise to a slowly migrating band on the gel that was difficult to interpret (*Figure 1—figure supplement 1j,k*); this problem did not arise when the NTD (GlpG residues 1–60) was replaced by residues 1–58 of the LepB protein (*Figure 3a*), and the corresponding $f_{FL}$ values are shown in the FP ($N$ = 131–224). The LepB part contains an N-terminal, $N_{out}$-$C_{in}$-oriented TMH (*Wolfe et al., 1983*; *von Heijne, 1989*), that interacts with the signal recognition particle Ffh (*Schibich et al., 2016*) and hence targets the LepB-GlpG constructs to the SecYEG translocon before GlpG TMH1 is translated. This could in principle affect the FP; however, because the C-terminal end of the LepB part is ≥70 residues away from the C-terminal end of the SecM AP in

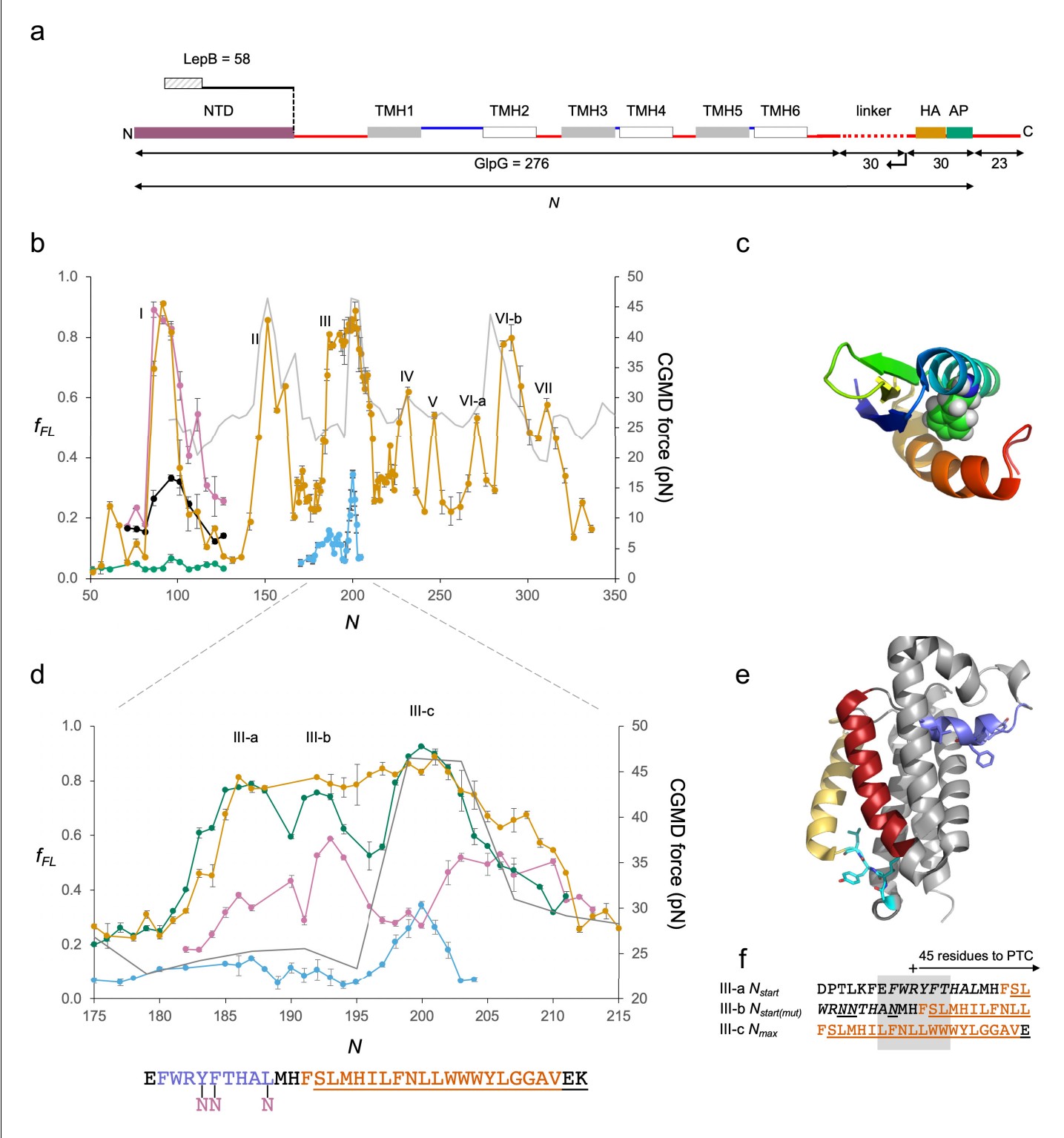

**Figure 3.** GplG. (**a**) Construct design, c.f., *Figure 2a*. The N-terminal LepB fusion is indicated. (**b**) Force profiles (FPs) for GlpG and LepB-GlpG ($N$ = 131–224) (orange), NTD($F^{16}$E) (green), in vitro translated N-terminal domain (NTD) (magenta), and NTD($F^{16}$E) (black), LepB-GlpG with SecM(*Ec*-Sup1) AP (blue), and coarse-grained molecular dynamics (CGMD)-FP calculated with a −100 mV membrane potential (gray). Error bars indicate SEM values. Note that the LepB-GlpG constructs are two residues shorter than the corresponding GlpG constructs but are plotted with the same $N$ values as the latter to facilitate comparison. (**c**) NTD (PDB ID: 2LEP), with $F^{16}$ in spacefill. (**d**) Enlarged FPs for LepB-GlpG with SecM(*Ec*) AP (orange), SecM(*Ec-Ms*) AP (green), SecM(*Ec*-sup1) AP (blue), and GlpG($Y^{138}F^{139}L^{143}$→NNN) with SecM(*Ec-Ms*) AP (magenta). CGMD-FP in gray. (**e**) Structure of GlpG with the

*Figure 3 continued on next page*

*Figure 3 continued*

periplasmic surface helix in blue, TMH2 in red, the membrane-associated cytoplasmic segment in cyan, and TMH5 in yellow. $Y^{138}F^{139}L^{143}$ and $G^{222}I^{223}Y^{224}L^{225}$ are shown as sticks. (f) LepB-GlpG peak III-a and III-c sequences aligned, respectively, from their $N_{start}$ and $N_{max}$ values, and the mutant LepB-GlpG($Y^{138}F^{139}L^{143}$→NNN) peak III-c sequence aligned from its $N_{max}$ value. Hydrophobic transmembrane helix (TMH) segments are shown in orange and transmembrane α-helices (PDB: 2IC8)underlined. The periplasmic surface helix is italicized. AP: arrest peptide; PTC: polypeptide transferase center.

The online version of this article includes the following figure supplement(s) for figure 3:

**Figure supplement 1.** GlpG.

these constructs, LepB is far outside the ribosome exit tunnel and therefore unlikely to exert a strong effect. Indeed, $f_{FL}$ values for GlpG (calculated either including or excluding the slowly migrating band in $I_{FL}$) and LepB-GlpG are very similar in the peak III region ($N$ = 166–231) of the FP (*Figure 3— figure supplement 1a*). $N_{start}$ and $N_{end}$ values for peaks II–VII are indicated in *Figure 3—figure supplement 1c,d*.

Peak I, at $N_{start}$ ≈ 84 residues, is conspicuously close to what would be expected for the folding of the NTD from previous studies of cotranslational folding of small globular domains in the ribosome exit tunnel (*Farías-Rico et al., 2018*). To verify that the peak indeed represents folding of the NTD, we recorded an FP for the NTD by in vitro transcription-translation in the PURE system (*Shimizu et al., 2005*) and further made a destabilizing point mutation ($F^{16}E$) in the core of the NTD (*Figure 3c*). The FP obtained in vitro (magenta) overlaps peak I in the in vivo FP, and the mutation strongly reduces $f_{FL}$ values for peak I both in vivo (green) and in vitro (black). Given that the NTD has a relative contact order of 15% and is predicted to fold on the ms time scale (*Plaxco et al., 1998*) while the elongation cycle on the ribosome takes ~100 ms/codon (*Young and Bremer, 1976*), the NTD has ample time to equilibrate between the unfolded and accessible folded states at each elongation step (*Kemp et al., 2019*). We conclude that the ~60 residue NTD folds inside the ribosome exit tunnel when its C-terminal end is 25–30 residues from the PTC, well before synthesis of the membrane domain has commenced.

Peaks II–VII in the FP correspond reasonably well to the CGMD-FP (gray) and HP (*Figure 3—figure supplement 1b*). The unexpectedly low $N_{start}$ value for peak III seems to be caused by an upstream periplasmic surface helix (*Figure 3f*) (see below). Likewise, peak VI-a likely reflects the membrane integration of a hydrophobic, membrane-associated cytoplasmic segment located just upstream of TMH5 (*Figure 3—figure supplement 1c*). In contrast, the unexpectedly high $N_{start}$ value for peak IV indicates that integration of TMH3 commences only when its N-terminal end is ~52 residues away from the PTC, possibly because of the tight spacing between TMH2 and TMH3.

As peak III saturates at $f_{FL}$ ≈ 0.9 over a rather wide range, we sought a more detailed view by using the strong SecM(*Ec*-Sup1) AP (*Yap and Bernstein, 2009*) (*Figure 3b,d*, blue) and the medium-strong SecM(*Ec-Ms*) AP (*Farías-Rico et al., 2017*) (*Figure 3d*, green). The SecM(*Ec*-Sup1) FP allows a precise determination of $N_{max}$ = 200, at which point the middle of TMH2 ($L^{155}$) is located 45 residues from the PTC (*Figure 3f*). The SecM(*Ec-Ms*) FP reveals additional detail: peak III is now seen to be composed of three subpeaks III-a, III-b, and III-c. III-a has $N_{start}$ = 182, coinciding with the N-terminal end of the periplasmic surface helix reaching 45 resides away from the PTC. For III-b, $N_{start}$ ≈ 190, with the N-terminal end of TMH2 ~45 residues from the PTC. The major subpeak III-c at $N$ ≈ 197–204 finally corresponds well to the peak seen in the SecM(*Ec*-Sup1) and the CGMD FPs, and therefore represents the membrane insertion of the most hydrophobic part of TMH2. Taken together, subpeaks III-b and III-c are reminiscent of the biphasic pulling force pattern previously recorded for a model hydrophobic transmembrane segment using the medium-strong SecM(*Ms*) AP (*Ismail et al., 2012*), which is closely related to the SecM(*Ec-Ms*) AP used here.

We further recorded a SecM(*Ec-Ms*) FP (magenta) for the triple mutation $Y^{138}F^{139}L^{143}$→NNN (*Figure 3e*) that renders the periplasmic surface helix less hydrophobic: the mutation strongly reduces the amplitude of peak III-a, has only a small effect on peak III-b, and both reduces the amplitude and shifts $N_{start}$ and $N_{max}$ for peak III-c by approximately four residues (*Figure 3d,f*). Thus, the periplasmic surface helix engages in hydrophobic interactions already during its passage through the translocon, presumably by sliding along a partly open lateral gate (*Cymer et al., 2015b*). It also adds to the force generated by the membrane integration of TMH2, possibly by partitioning into the

periplasmic leaflet of the inner membrane at approximately the same time that TMH2 enters the translocon.

## BtuC: 10 TMHs, 326 residues

Finally, we studied BtuC, a vitamin B12 transporter with 10 TMHs, as an example of a large, multi-spanning protein with a complex fold (*Hvorup et al., 2007*). In order to improve expression, we added the N-terminal part of LepB to the BtuC constructs (*Figure 4a*) and used a LepB antiserum for immunoprecipitation. The $N_{out}$-$C_{in}$ orientation of LepB TMH1 ensures that the $N_{in}$-$C_{in}$ topology of BtuC will be maintained, and constructs that we could measure without the LepB fusion gave similar $f_{FL}$ values as those seen for the LepB fusions (*Figure 4—figure supplement 1b*).

We identified 11 peaks in the FP (*Figure 4b*, orange), one more than could be accounted for by the 10 TMHs. Since it was not possible to provide an unequivocal match between the BtuC FP and the CGMD-FP (or HP, *Figure 4—figure supplement 1a*), we did two sets of controls. First, we chose constructs at or near peaks in the FP and CGMD-FP and mutated multiple hydrophobic residues (Leu, Ile, Val, Met) located 40–50 residues from the PTC to less hydrophobic Ala residues (*Figure 4—figure supplement 2*). The mutations caused significant drops in $f_{FL}$ (p < 0.01, two-sided t-test), except for construct $N$ = 191 that is mutated at the extreme N-terminus of TMH5. The mutation data allowed us to identify the membrane integration of TMHs 1, 2, 3, 4, 5, 7, 8, 9, and 10 with peaks I, II, III, IV, V, VIII, IX, X, and XI, respectively; the overlapping peaks VIII and IX appear to represent the concerted integration of the closely spaced TMH7 and TMH8. However, peak II (corresponding to TMH2) is shifted to unexpectedly high, and peaks V (corresponding to TMH5), X (corresponding to TMH9), and XI (corresponding to TMH10) to unexpectedly low, $N_{start}$ values (*Figure 4c*). To confirm these assignments, we obtained FPs for the isolated TMH2 (dashed green), TMH8 (dashed light blue), and TMH10 (dashed pink) sequences (*Figure 4b*) by introducing them into the periplasmic domain of LepB such that they maintained their natural $N_{out}$-$C_{in}$ orientation (*Figure 4d*); the FPs for the individual TMHs overlap the corresponding peaks II, IX, and XI in the full FP. Likewise, an FP obtained for a construct lacking TMH1-TMH4 overlaps the full FP, except that peak V is shifted to a higher $N_{start}$ value (*Figure 4—figure supplement 3*), more in line with the peak seen in the CGMD-FP. The low $N_{start}$ value for the $N_{in}$-$C_{out}$-oriented TMH5 in full-length BtuC may result from an early interaction between a positively charged patch (RFARRHLSTSR) just upstream of TMH5 and negatively charged lipid headgroups (note that only two of the four Arg residues are present in the ΔTMH1-TMH4 construct; *Figure 4—figure supplement 3*), while the low $N_{start}$ values for peaks X and XI are likely caused by the short upstream hydrophobic segments LCGL and LAAALEL (*Figure 4c,f*), similar to peak III in GlpG.

Remarkably, the N-terminal end of the isolated TMH2 is ~45 residues away from the PTC at $N_{start}$, suggesting that upstream sequence elements present in full-length BtuC delay the integration of TMH2 by ~10 residues (compare II* and II in *Figure 4c*). The most conspicuous feature in the upstream region of TMH2 is the presence of three positively charged Arg residues, an uncommon occurrence in a periplasmic loop (*Heijne, 1986*). Indeed, when these residues are replaced by uncharged Gln residues in LepB-BtuC, peak II (dashed black in *Figure 4b,e*) becomes almost identical to the FP for the isolated TMH2; a similar behavior is seen when the CGMD-FP simulation is run without an electrical membrane potential (*Figure 4e*). Upstream positively charged residues thus delay the membrane integration of the $N_{out}$-oriented TMH2, possibly because of the energetic cost of translocating them against the membrane potential (*Ismail et al., 2015*), or because they are temporarily retained in the negatively charged exit tunnel (*Mercier et al., 2020*).

Neither peak VI nor VII seems to represent the integration of TMH6, but instead flanks the location expected from the CGMD-FP and HP and apparently corresponds, respectively, to the membrane insertion of a short periplasmic re-entrant helix and of a short cytoplasmic surface helix (*Figure 4c,h*). Mutation of three hydrophobic residues to Ala in the latter significantly reduces the amplitude of peak VII (*Figure 4—figure supplement 2*, construct $N$ = 259). Further, the FP for the isolated TMH6 (*Figure 4b,g*, dashed dark blue) peaks in the location expected from the CGMD-FP, between peaks VI and VII, and the FP for the isolated TMH5-6 part that includes the re-entrant helix but lacks the downstream surface helix is intermediate between the LepB-BtuC and the TMH6 FPs (*Figure 4g*, dashed green). Thus, the membrane interactions of the periplasmic re-entrant helix and the cytoplasmic surface helix exert a strong effect on the membrane integration of the intervening TMH6.

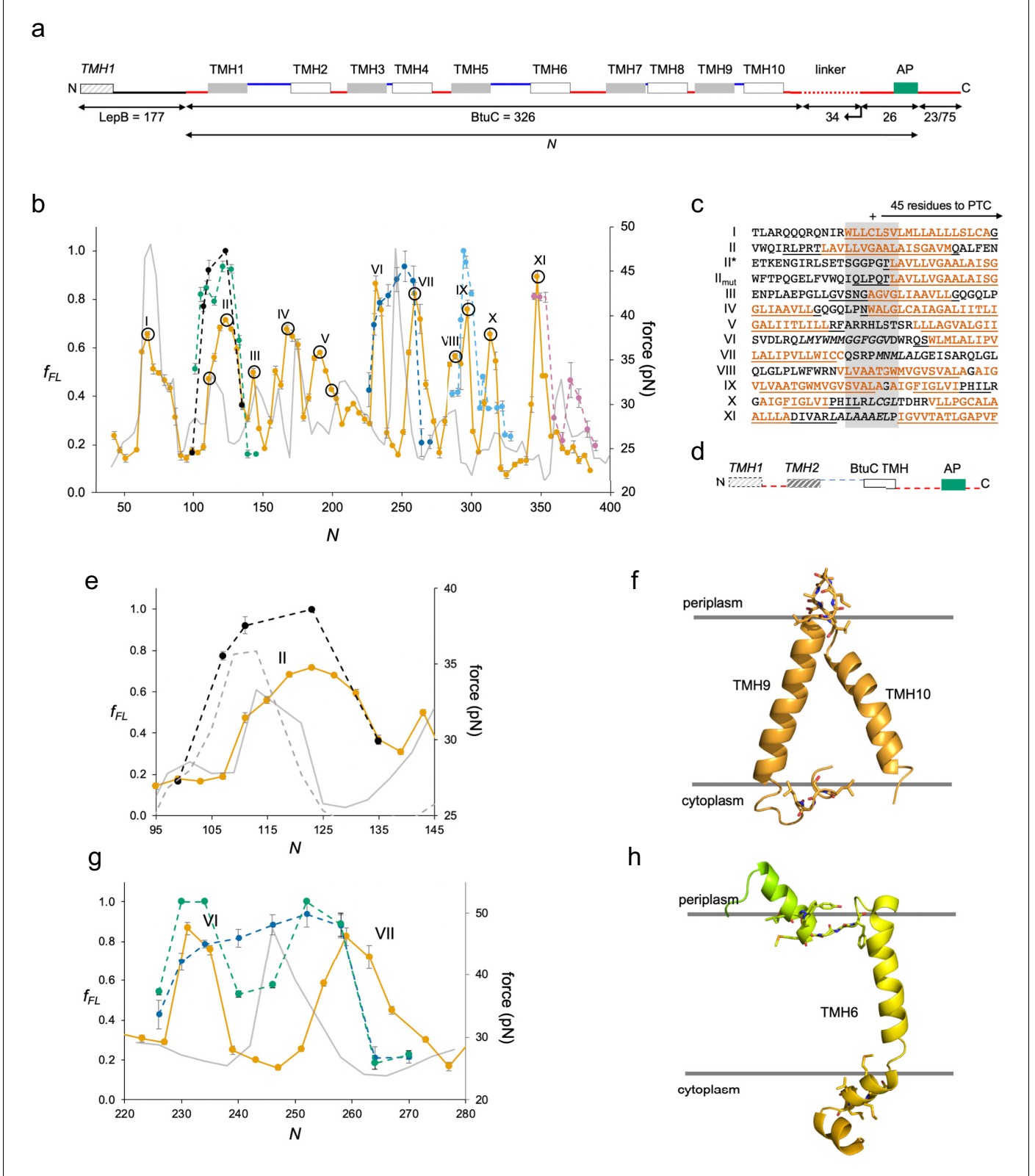

**Figure 4.** BtuC. (a) Construct design, cf. *Figure 2a*. The N-terminal LepB fusion is indicated. *N* values are calculated from the N-terminus of BtuC. For constructs with *N* ≥ 298, the C-terminal tail is 75 residues long. Circles indicate constructs for which mutations were made in the corresponding transmembrane helix (TMH) (see *Figure 4—figure supplement 2*. (b) Force profiles (FPs) for BtuC (orange), BtuC-TMH2 (green), BtuC (R⁴⁷R⁵⁶R⁵⁹→QQQ) (black), BtuC-TMH6 (dark blue), BtuC-TMH8 (blue), BtuC-TMH10 (pink), and CGMD-FP calculated with a −100 mV membrane

*Figure 4 continued on next page*

*Figure 4 continued*

potential (gray). Error bars indicate SEM values. Note that the BtuC-TMH2, BtuC-TMH6, BtuC-TMH8, and BtuC-TMH10 constructs are plotted with the same $N$ values as the corresponding BtuC constructs to facilitate comparison (i.e., the number of residues between the TMH in question and the last residue of the AP is the same in both types of constructs, see ***Supplementary file 1***). (c) Sequences corresponding to peaks I–XI aligned from their $N_{start}$ values. Hydrophobic TMH segments are shown in orange and membrane-embedded α-helices according to the OPM database (*Lomize et al., 2012*) underlined. Re-entrant loops and surface helices discussed in the text are italicized. (d) Construct design for obtaining FPs of isolated $N_{out}$-oriented BtuC TMHs. Dashed segments are derived from LepB. (e) Enlarged FPs for BtuC (orange) and (R⁴⁷R⁵⁶R⁵⁹→QQQ) (black), together with coarse-grained molecular dynamics (CGMD)-FPs calculated with (gray) and without (dashed gray) a −100 mV potential. (f) BtuC TMH9-TMH10, with hydrophobic flanking residues in stick representation (PDB ID: 2QI9). (g) Enlarged FPs for BtuC (orange), isolated TMH6 (residues 187–206; blue), and isolated TMH5-6 (residues 138–206; green). In the latter construct, LepB TMH2 was not included in order to maintain the correct membrane topology of the BtuC TMH5-TMH6 part. The CGMD-FP is in gray. (h) Structure of TMH6 including the upstream periplasmic re-entrant helix and the downstream cytoplasmic surface helix, with hydrophobic flanking residues in stick representation. AP: arrest peptide; PTC: polypeptide transferase center.

The online version of this article includes the following figure supplement(s) for figure 4:

**Figure supplement 1.** BtuC.
**Figure supplement 2.** Mutations in constructs representing peaks in the BtuC force profile (FP).
**Figure supplement 3.** BtuC.

## Discussion

A detailed view of the cotranslational integration of three multispanning membrane proteins provided here shows that translocating nascent chains experience a distinct transition to a more hydrophobic environment at a distance of ~45 residues from the PTC, generating an oscillating force on the nascent chain that is ultimately transmitted to the PTC and varies in step with the appearance of each TMH in the vicinity of the SecYEG translocon channel. It seems likely that such oscillations can have multiple effects on the translation of membrane proteins, as recently demonstrated for ribosomal frameshifting (*Harrington et al., 2020*), and may affect protein quality control (*Lakshminarayan et al., 2020*).

Notably, TMHs also stop generating a force on the nascent chain when their C-terminal end reaches ~45 residues from the PTC, irrespective of whether their orientation is $N_{out}$-$C_{in}$ or $N_{in}$-$C_{out}$. This is in agreement with the 'sliding' model of TMH integration (*Cymer et al., 2015b*), which posits that $N_{out}$-$C_{in}$ TMHs have continuous lipid contact as they slide across the membrane along the open lateral gate in the SecYEG translocon, while $N_{in}$-$C_{out}$ TMHs first partition into the cytoplasmic interface region of the membrane as they exit the ribosome (and therefore generate less pulling force than $N_{out}$-$C_{in}$ TMHs (*Cymer et al., 2014*) and only insert across the membrane as their polar C-terminal flanking region translocates through the central translocon channel. In both cases, the TMHs are embedded in the membrane (albeit in perpendicular orientations) when their C-terminal end is ~45 residues from the PTC. In the sliding model, the translocon channel serves as a conduit for polar nascent chain segments while hydrophobic segments are always in contact with surrounding lipid, similar to what has been proposed for the YidC/Oxa1 translocon family (*He et al., 2020*). The lateral gate region in the SecYEG translocon thus in a certain sense mimics the water–bilayer interface environment (*Marx and Fleming, 2021*).

We also find that the cytoplasmic NTD in GlpG folds already in the ribosome exit tunnel, before the first TMH has been synthesized. Further, the FPs for EmrE, GlpG, and BtuC to a first approximation match those predicted by CGMD calculations, but uncover a much richer picture of the membrane integration process where charged residues and membrane-interacting segments such as re-entrant loops and surface helices flanking a TMH show prominent interactions with the translocon and surrounding lipid. Finally, point mutations in EmrE TMH1 affect the pulling force generated by downstream TMHs in a highly position-dependent manner, suggestive of residue-specific interactions between TMHs during the membrane integration process. Complementing in vitro unfolding/folding studies (*Yu et al., 2017*; *Choi et al., 2019*), real-time FRET analyses (*Mercier et al., 2020*), chemical crosslinking (*He et al., 2020*), structure determination (*Kater et al., 2019*), and computational modeling (*Lu et al., 2018*), high-resolution in vivo FPA can thus help identify the molecular interactions underlying cotranslational membrane protein biogenesis with up to single-residue precision.

# Materials and methods

## Key resources table

| Reagent type (species) or resource | Designation | Source or reference | Identifiers | Additional information |
|---|---|---|---|---|
| Strain, strain background (*Escherichia coli*) | BL21(DE3) | Sigma-Aldrich | CMC0016 | Electrocompetent cells |
| Strain, strain background (*Escherichia coli*) | MC1061 | J Biol Chem. 261:13844–9. PMID:3531212 | NA | Electrocompetent cells |
| Other | Protein-G-agarose | Roche | 11243233001 | Resin used for immunoprecipitation |
| Antibody | Anti-HA.11 epitope tag antibody (mouse monoclonal) IgG | BioLegend | Cat# 901533 | Used for immunoprecipitation (1 μl of 1 mg/ml, diluted 1:820) |
| Antibody | LepB antibody (rabbit polyclonal) IgG | Generated in-house | NA | Used for immunoprecipitation (dilution 1:820) |
| Recombinant DNA reagent | pET Duet-1 (plasmid) | Novagen | Cat# 71146 | Expression plasmid |
| Recombinant DNA reagent | pING1 (plasmid) | Gene 34:137–45. PMID:4007491 | NA | Expression plasmid |
| Commercial assay, kit | GeneJET Plasmid miniprep kit | Thermo Fisher Scientific RRID: SCR_008452 | Cat# 0502 | Used to purify plasmids |
| Commercial assay, kit | GeneJET PCR Purification Kit | Thermo Fisher Scientific | Cat# K0701 | Used to purify linear fragments for in vitro expression |
| Commercial assay, kit | PURExpress | New England Biolabs | Cat# E6800L | Used for in vitro expression |
| Chemical compound, drug | $^{35}$S methionine | PerkinElmer | Cat# NEG009T001MC | $^{35}$S Methionine is incorporated into the protein during in vitro and in vivo translation and aids detection by phosphor imaging |
| Software, algorithm | EasyQuant | Developed in-house Nat Struct Mol Biol. 19:1018–22. PMID: 23001004 | | Used to quantify relative fraction full length of translated protein from SDS-PAGE |

## Enzymes and chemicals

All enzymes used in this study were purchased from Thermo Fisher Scientific (USA) and New England Biolabs (USA). Oligonucleotides were from Eurofins Genomics (Germany). DNA isolation/purification kits and precast polyacrylamide gels were from Thermo Fisher Scientific (USA). L-[$^{35}$S]-methionine was obtained from PerkinElmer (USA). Mouse monoclonal antibody against the HA antigen was purchased from BioLegend (USA). Protein G-agarose beads were manufactured by Roche (Switzerland). All other reagents were from Sigma-Aldrich (Germany).

## Cloning and mutagenesis

### EmrE

The previously described $N_{out}$-$C_{out}$-oriented EmrE($C_{out}$) version carrying mutations $T^{28}R$, $L^{85}R$, and $R^{106}A$ was engineered in a pETDuet-1 vector (*Rapp et al., 2007*). A series of constructs was designed by inserting nucleotides downstream of EmrE($C_{out}$) coding for a variable LepB-derived linker sequence (between 4 and 34 residues), the 9-residue long HA tag, the 17-residue long *E. coli* SecM AP, and a 23-residue long C-terminal tail. The following APs with stalling strengths were used: SecM(*Ec*) (FSTPVWISQAQGIRAGP), SecM(*Ec-Ms*) (FSTPVWISQHAPIRGSP, mutations underlined),

and SecM(*Ec*-Sup1) (FSTPVWISQA<u>PPI</u>RAGP, mutations underlined). The LepB-derived linker as well as EmrE(C$_{out}$) were truncated 2–5 residues at a time from the C-terminus of the respective sequence. Site-specific DNA mutagenesis was carried out to introduce point mutations E$^{14}$L, E$^{14}$A, E$^{14}$D, and E$^{14}$Q in EmrE(C$_{out}$). All cloning and mutagenesis products were confirmed by DNA sequencing. Different EmrE sequences used in this study are summarized in *Supplementary file 1*.

## GlpG

The gene encoding for GlpG was amplified from the genome of *E. coli* K-12 MG1655 strain by PCR and assembled together with other sequence elements into the pING1 plasmid (*Johnston et al., 1985*; *Dalbey and Wickner, 1985*) by Gibson assembly (in-house). For the longest truncates, a LepB-derived unstructured linker was introduced downstream of the GlpG sequence, followed by an HA tag, a 17-residue long *E. coli* SecM AP, and a 23-residue long C-terminal tail derived from LepB. Partially overlapping primers were used in around-the-horn PCR (*Floor, 2018*) to create deletion variants truncating upstream of the HA tag. All the sequences of GlpG deletion variants used in this study are summarized in *Supplementary file 1*. For LepB-GlpG constructs, 60 N-terminal residues corresponding to the soluble domain were truncated from GlpG and substituted by LepB N-terminal segment comprising TMH1 and a long cytoplasmic loop (1–174 res of LepB). Three different stalling sequences of increasing strength were used: SecM(*Ec*) (FSTPVWISQAQGIRAGP), SecM(*Ec-Ms*) (FSTPVWISQ<u>HAPIR</u><u>GS</u>P, mutations underlined), and SecM(*Ec*-Sup1) (FSTPVWISQA<u>PPI</u>RAGP, mutations underlined). Mutations in SecM(*Ec*) AP and GlpG folding variants NTD(F$^{16}$E), GlpG (Y$^{138}$F$^{139}$L$^{143}$→NNN) were engineered using partially overlapping primers in around-the-horn PCR. All cloning and mutagenesis products were confirmed by DNA sequencing.

For in vitro transcription/translation of the soluble NTD domain, constructs of variable length were fused to the SecM(*Ec*) AP and cloned into the pET19b vector. Folding variant NTD(F$^{16}$E) was engineered using partially overlapping primers in around-the-horn PCR. pET19b plasmids containing different GlpG variants were used as template to create linear DNA fragments amplified by PCR for each construct using forward and reverse primers that anneal to the T7 promoter and terminator regions, respectively.

## BtuC

The previously described pING1 plasmid harboring a truncated LepB sequence with an inserted hydrophobic test segment (6L/13A) followed by a variable LepB-derived linker (between 9 and 43 residues), the 17-residue long *E. coli* SecM AP, and a C-terminal tail comprising 23 or 75 residues derived from LepB was used to generate all BtuC constructs (*Ismail et al., 2012*). All BtuC sequences used in this study are summarized in *Supplementary file 1*. The gene encoding BtuC was amplified from the genome of the *E. coli* K-12 MG1655 strain by PCR and then engineered to replace 6L/13A using Gibson assembly (in-house) (*Gibson et al., 2009*). In order to maintain the correct topology of BtuC, the sequence coding for TMH2 of LepB (between residues P$^{58}$ and P$^{114}$) was removed by deletion-PCR, resulting in a 177-residue-long sequence upstream of BtuC. A gene sequence encoding 52 residues (part of LepB P2 domain) was introduced downstream of the SecM AP for constructs with $N \geq 298$, resulting in an extension of the C-terminal tail from 23 to 75 residues in order to improve protein separation during SDS-PAGE. The LepB-derived linker as well as BtuC were truncated four residues at a time from the C terminus of the respective sequence. Site-specific DNA mutagenesis was carried out to replace three or six hydrophobic residues with Ala residues in TMHs of BtuC and to replace three Arg residues with Gln residues in the periplasmic loop connecting TMH1 and TMH2 (R$^{47}$R$^{56}$R$^{59}$→QQQ). Gene sequences of single TMH and 2-TMH constructs were cloned with the variable linker sequence derived from LepB, and the single TMH constructs were placed in the background containing gene sequences of both LepB TMHs in order to maintain the correct topology. Furthermore, BtuCΔLepB constructs lacking the N-terminal LepB fusion were obtained by deletion of the entire LepB sequence upstream of BtuC, and the 9-residue long LepB-derived linker was replaced with an HA tag for immunoprecipitation. All cloning and mutagenesis products were confirmed by DNA sequencing.

## In vivo pulse-labeling analysis

Competent *E. coli* MC1061 (*Dalbey and Wickner, 1986*) or BL21 (DE3) cells were transformed with the respective pING1 (BtuC, GlpG) or pET Duet-1 (EmrE) plasmid, respectively, and grown overnight at 37°C in M9 minimal medium supplemented with 19 amino acids (1 µg/ml, no Met), 100 µg/ml thiamine, 0.4% (w/v) fructose, 100 mg/ml ampicillin, 2 mM $MgSO_4$, and 0.1 mM $CaCl_2$. Cells were diluted into fresh M9 medium to an $OD_{600}$ of 0.1 and grown until an $OD_{600}$ of 0.3–0.5. Expression from pING1 was induced with 0.2% (w/v) arabinose and continued for 5 min at 37°C. Expression from pET Duet-1 was induced with 1 mM IPTG and continued for 10 min at 37°C. Proteins were then radiolabeled with [$^{35}$S]-methionine for 2 min (1 min for BtuC constructs lacking the N-terminal LepB fusion) at 37°C before the reaction was stopped by adding ice-cold trichloroacetic acid (TCA) to a final concentration of 10%. Samples were put on ice for 30 min and precipitates were spun down for 10 min at 20,000 *g* at 4°C in a tabletop centrifuge (Eppendorf, Germany). After one wash with ice-cold acetone, centrifugation was repeated and pellets were subsequently solubilized in Tris-SDS buffer (10 mM Tris-Cl pH 7.5, 2% [w/v] SDS) for 5 min while shaking at 1400 rpm at 37°C. Samples were centrifuged for 5 min at 20,000 *g* to remove insoluble material. The supernatant was then added to a buffer containing 50 mM Tris-HCl pH 8.0, 150 mM NaCl, 0.1 mM EDTA-KOH, 2% (v/v) triton X-100, and supplemented with Pansorbin (Sigma-Aldrich) (BtuC constructs) or Protein-G-agarose (Roche) (all GlpG and EmrE constructs, and BtuC constructs lacking the N-terminal LepB fusion). After 15 min incubation on ice, non-specifically bound proteins were removed by centrifugation at $20,000 \times g$ (when Pansorbin was used) or $7000 \times g$ (when Protein-G-agarose was used). The supernatant was used for immunoprecipitation of BtuC constructs using Pansorbin and LepB antisera (rabbit) (in-house), or immunoprecipitation of GlpG/EmrE constructs using Protein-G-agarose and Anti-HA.11 Epitope Tag Antibody (mouse) (BioLegend). The incubation was carried out at 4°C while rolling. After centrifugation for 1 min, immunoprecipitates were washed with 10 mM Tris-Cl pH 7.5, 150 mM NaCl, 2 mM EDTA, and 0.2% (v/v) triton X-100 and subsequently with 10 mM Tris-Cl pH 7.5. Samples were spun down again and pellets were solubilized in SDS sample buffer (67 mM Tris, 33% [w/v] SDS, 0.012% [w/v] bromophenol blue, 10 mM EDTA-KOH pH 8.0, 6.75% [v/v] glycerol, 100 mM DTT) for 10 min while shaking at 1400 rpm. Solubilized proteins were incubated with 0.25 mg/ml RNase for 30 min at 37°C and subsequently separated by SDS-PAGE on Bis-Tris gels (Thermo Fisher Scientific). Gels were fixed in 30% (v/v) methanol and 10% (v/v) acetic acid and dried by using a Bio-Rad gel dryer model 583 (Bio-Rad Laboratories, US). Radiolabeled proteins were detected by exposing dried gels to phosphorimaging plates, which were scanned in a Fujifilm FLA-3000 scanner (Fujifilm, Japan). Band intensity profiles were obtained using the FIJI (ImageJ) software and quantified with our in-house software EasyQuant. $A_c$ and/or $FL_c$ controls were included in the SDS-PAGE analysis for constructs where the identities of the $A$ and $FL$ bands were not immediately obvious on the gel. Data was generally collected from three independent biological replicates for EmrE and BtuC, and for two or three replicates for GlpG, and averages and SEM were calculated (see *Source data 1*). Note that for two replicates plotting the average ± SEM is equivalent to plotting the average ± error bars representing the two experimental measurements.

## In vitro transcription/translation of GlpG NTD

In vitro transcription/translation was performed using the commercially available PURExpress system (New England Biolabs). Reactions were mixed according to the manufacturer's recommendations by the addition of 2.2 µl of linear DNA of each construct giving a final volume of 10 µl. Polypeptide synthesis was carried out in the presence of [$^{35}$S]-methionine at 37°C for 15 min under 700 rpm shaking. Translation was stopped by the addition of TCA to a final concentration of 5% and incubated on ice for at least 30 min. Total protein was sedimented by centrifugation at 20,000 *g* for 10 min at 4°C in a tabletop centrifuge (Eppendorf, Germany). The pellet was resuspended in 2× SDS/PAGE sample buffer, supplemented with RNaseA (400 µg/ml) to digest the stalled peptidyl-tRNA, and incubated at 37°C for 15 min under 1000 rpm agitation. The samples were resolved on 12% Bis-Tris gels (Thermo Fisher Scientific) in MOPS buffer. Gels were dried on Hoefer GD 2000 dryer (Hoefer, US), exposed to a phosphorimager screen for 24 hr, and scanned using the Fujifilm FLA-9000 phosphorimager for visualization of radioactively labeled protein species.

## Molecular dynamics simulations

Computer simulations of cotranslational membrane integration were carried out using a previously developed and validated CGMD model in which nascent proteins are mapped onto CG beads representing three amino acids (*Niesen et al., 2017*; *Niesen et al., 2018*). The nascent protein interacts with the Sec translocon and the ribosome via pairwise interactions that depend on the hydrophobicity and charge of the beads of the nascent protein. The interaction parameters are unchanged from previous work (*Niesen et al., 2017*). The lateral gate of the translocon switches between the open and closed conformations with probability dependent on the difference in free energy between the two conformations. The structures of the ribosome and translocon are based on cryo-EM structures and, aside from the lateral gate of the translocon, are fixed in place during the simulations. The lipid bilayer and cytosol are modeled implicitly. The positions of the nascent protein beads are evolved using overdamped Langevin dynamics with a timestep of 300 ns and a diffusion coefficient of 253 nm$^2$/s. Membrane potentials are included by adding an electrostatic energy term to the simulations, as previously described (*Niesen et al., 2018*).

To simulate protein translation, new amino acids are added to the nascent chain at a rate of five amino acids per second. Simulations of EmrE, GlpG, and BtuC begin with 12 amino acids translated. Translation continues until the nascent protein reaches the desired length, at which point translation is halted and forces on the C-terminus of the nascent chain are measured every 3 ms for 6 s. This methodology has been found to accurately reproduce experimental FPs (*Niesen et al., 2018*). Forces are measured starting at a nascent protein length of 18 amino acids for EmrE and BtuC, and 70 for GlpG. The computational force profile (CGMD-FP) is then obtained by measuring the forces at lengths incremented by four amino acids. Simulations at different lengths are performed independently and repeated 100 times. Because the ribosomal exit tunnel is truncated in the CGMD model, a shift in the protein index is required to compare simulated and experimental results. Shifts of −12, –5, and −5 residues are used for EmrE, GlpG, and BtuC CGMD-FPs, respectively. The shifts are estimated by aligning the computational and experimental FPs and are in line with what is expected given the length of the truncated exit tunnel. Variation in the shift may reflect different degrees of compaction of the nascent chain. Although previous work provides a framework to estimate the experimentally observed fraction full length from simulated forces given a specific AP (*Niesen et al., 2018*; *Tian et al., 2018*), forces are reported directly to facilitate comparison between experiments performed with different APs.

## Protein contact order calculation

The relative contact order for the GlpG NTD was calculated using the Contact Order server at https://depts.washington.edu/bakerpg/contact_order/contact_order.cgi.

## Acknowledgements

We thank Dr. Rickard Hedman (Stockholm University) for programming and maintenance of the EasyQuant software. This work was supported by grants from the Knut and Alice Wallenberg Foundation (2017.0323), the Novo Nordisk Fund (NNF18OC0032828), and the Swedish Research Council (621-2014-3713) to GvH, from a Marie Curie Initial Training Network Grant (Horizon 2020, Protein-Factory 642863) to FN, and from NIGMS, National Institutes of Health, (R01GM125063) to TFM and MZ. This work used the Extreme Science and Engineering Discovery Environment (XSEDE) Bridges computer at PSC through allocation TG-MCB160013. XSEDE is supported by National Science Foundation grant number ACI-1548562.

## Additional information

### Funding

| Funder | Grant reference number | Author |
|---|---|---|
| Knut och Alice Wallenbergs Stiftelse | 2017.0323 | Gunnar von Heijne |
| Novo Nordisk Fonden | NNF18OC0032828 | Gunnar von Heijne |

| Vetenskapsrådet | 621-2014-3713 | Gunnar von Heijne |
|---|---|---|
| National Institutes of Health | R01GM125063 | Matthew Zimmer<br>Thomas F Miller III |
| Horizon 2020 Framework Programme | Protein Factory 642863 | Felix Nicolaus |
| National Science Foundation | ACI-1548562 | Thomas F Miller III |

The funders had no role in study design, data collection and interpretation, or the decision to submit the work for publication.

**Author contributions**
Felix Nicolaus, Ane Metola, Daphne Mermans, Formal analysis, Supervision, Investigation, Visualization, Methodology, Writing - review and editing; Amanda Liljenström, Ajda Krč, Salmo Mohammed Abdullahi, Investigation; Matthew Zimmer, Formal analysis, Funding acquisition, Investigation, Writing - review and editing; Thomas F Miller III, Conceptualization, Funding acquisition, Writing - review and editing; Gunnar von Heijne, Conceptualization, Resources, Formal analysis, Supervision, Funding acquisition, Validation, Visualization, Methodology, Writing - original draft, Project administration, Writing - review and editing

**Author ORCIDs**
Felix Nicolaus  https://orcid.org/0000-0001-9230-8544
Ane Metola  https://orcid.org/0000-0002-2885-7634
Daphne Mermans  https://orcid.org/0000-0001-6001-5608
Matthew Zimmer  https://orcid.org/0000-0002-1437-2636
Thomas F Miller III  https://orcid.org/0000-0002-1882-5380
Gunnar von Heijne  https://orcid.org/0000-0002-4490-8569

**Decision letter and Author response**
Decision letter https://doi.org/10.7554/eLife.64302.sa1
Author response https://doi.org/10.7554/eLife.64302.sa2

# Additional files

## Supplementary files
• Source data 1. Measured fFL values for all EmrE, GlpG, and BtuC constucts reported in *Figures 2, 3* and *4* (and the corresponding Supplementary Figures and *Supplementary file 1*).

• Supplementary file 1. Amino acid sequences of EmrE($C_{out}$), GlpG, and BtuC constructs.

• Transparent reporting form

## Data availability
All fFL values measured in this study are included as figures Source Data.

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
