## [Decision Letter]

**Acceptance summary:**

This is an elegant report on the ubiquitous process of membrane protein insertion, studied in vivo and in vitro using the bacterial model *E. coli* – exploiting three exemplar membrane proteins of increasing topological complexity. The exploitation of force profile analysis (FPA) developed in the von Heijne lab – now at high resolution – has proved very powerful and complementary to other techniques; such as MD simulations (also deployed here). The outcome is new knowledge about the molecular details of membrane protein insertion. Importantly the paper reveals a number of very interesting aspects of this process.

**Decision letter after peer review:**

Thank you for submitting your article "Residue-by-residue analysis of cotranslational membrane protein integration in vivo" for consideration by *eLife*. Your article has been reviewed by three peer reviewers, and the evaluation has been overseen by a Reviewing Editor and Olga Boudker as the Senior Editor. The following individual involved in review of your submission has agreed to reveal their identity: Ian Collinson (Reviewer #2).

The reviewers have discussed the reviews with one another and the Reviewing Editor has drafted this decision to help you prepare a revised submission.

This is an elegant report on the ubiquitous process of membrane protein insertion, studied in vivo and in vitro using the bacterial model *E. coli* – exploiting three exemplar membrane proteins of increasing topological complexity.

Our knowledge about the mechanism of membrane protein insertion is somewhat superficial and the authors are leading the way in this area. This manuscript is a prime example – and thus worthy of publication in close to its present form (certainly no more experiments required).

The exploitation of force profile analysis (FPA) developed in the von Heijne lab – now at high resolution – has proved very powerful and complementary to other techniques; such as MD simulations (also deployed here). The outcome is new knowledge about the molecular details of membrane protein insertion. Importantly the paper reveals a number of very interesting aspects of this process:

Summary:

This is an elegant report on the ubiquitous process of membrane protein insertion, studied in vivo and in vitro using the bacterial model *E. coli* – exploiting three exemplar membrane proteins of increasing topological complexity.

Our knowledge about the mechanism of membrane protein insertion is somewhat superficial and the authors are leading the way in this area. This manuscript is a prime example – and thus worthy of publication in close to its present form (certainly no more experiments required).

The exploitation of force profile analysis (FPA) developed in the von Heijne lab – now at high resolution – has proved very powerful and complementary to other techniques; such as MD simulations (also deployed here). The outcome is new knowledge about the molecular details of membrane protein insertion. Importantly the paper reveals a number of very interesting aspects of this process:

Revisions:

The following is excerpted from the consultation session which specifies the key points the reviewers feel need to be addressed in your revised submission. I have included the full comments of the reviewers for your consideration. However, these need not be addressed in detail in your the response accompanying the submission of your revised manuscript.

There are a number of technical issues that need to be addressed in a revised text including conclusions that need to be adjusted. If co-translational folding of polypeptide segments in the ribosome tunnel can affect the integration process (which the authors claim occurs), in some cases it is problematic to use Lep fusions to solve technical issues with detection. In those cases, the native polypeptide is no longer the context for this detection issue.

Perhaps the authors could show the force profiles are unaffected by the presence of the fusion proteins? Of course, if the results are affected then the conclusions should be changed accordingly.

However, it may be difficult to show this convincingly given the use of Lep with Btu to allow an interpretation of the protein band pattern. The implication is that Lep affects the process of either integration or folding. Further, the authors claim sequences upstream of the N-terminus may affect integration by co-translational folding as in the case of the NTD region in GlpG. Thus the replacement of Lep for such sequences may interfere with the folding/integration process.

Given the von Heijne group's previous good agreement with MD simulations, perhaps this could approach could be used to access the divergence from native behavior.

Perhaps the current analysis would be acceptable for the constructs that are sufficiently long (For GlpP, N>120; a 58-residue-long LepB portion is completely exposed outside the ribosome and does not engage with SecY). GlpP is probably fine in this regard because LepB-fused constructs are N=131-224. Of course, LepB has a membrane anchor, and we don't know how that might have affected FP of GlpP. They might have some native GlpG data besides those with multiple bands, which might be useful. For BtuC, they show at least some data without LepB fusion (Figure 4—figure supplement 1B). This is nice particularly because these are shorter constructs, which could have been more affected by LepB. Nevertheless, the potential caveats in their interpretations bear correction.

It may be helpful to include raw images and some clarification regarding statistics for the experiments where there were only 2 biological replicates..

The major issues summarized above may be addressed by the authors without additional experiments. Reviewer #1 offers additional experimental tests, but they are not considered essential for a satisfactory revision of this most interesting work.

Reviewer #1:

The manuscript by Nicolaus et al. followed the co-translational integration of three multi-spanning membrane proteins using force profile (FP) analysis, the method previously pioneered by the von Heijne group. This approach uses a C-terminal fusion of a translation arrest peptide to either soluble or membrane proteins of varying truncations. Folding or membrane integration of the nascent chain produces a pulling force, which can be read out from how efficiently the arrest peptide causes translational stalling. Previously the group has analyzed membrane insertion of model single transmembrane helices (TMHs) in detail (Ismail et al., 2012). Now they extended the approach to multi-spanning (and more native) membrane proteins, namely EmrE, GlpG, and BtuC. The authors concluded that their data support the so-called "sliding" model for TMH integration, where each TMH slides down along the open lateral gate of the SecYEG channel. They also present several interesting observations, such as effects of charged residues, re-entrant loops, and surface helices on the timings of TMS integration, although these would require more data to be generalized.

A main weakness of the study is its exclusive reliance on the FP analysis. Coarse-grained MD simulations were also used but rather to confirm observations from the FP analysis. A limitation of the FP analysis is that it only monitors (indirectly) an insertion of the last TMH which is near the SecYEG channel. The analysis does not address what would happen to earlier TMHs (such as folding of TMD, delayed insertion, or possible flipping of TMs) as the polypeptide grows. A second weakness would be that the study adds only rather minor updates (effects of charged residues, re-entrant loops, and surface helices) to the previous model.

Despite the weaknesses, this work provides useful insights and resources to the field, and therefore I consider it to be suitable for publication in *eLife* with revision. Membrane integration of multi-spanning membrane proteins is a still poorly understood process, largely because there are only very few experimental tools available for researchers to follow the process. The authors' analysis is generally of high quality, and their interpretations are reasonable.

1) The current version of the manuscript is probably difficult for scientists outside the field to follow. It would be better to extend Introduction to provide non-experts with more background information about the pathway, such as the general model for the SecYEG-mediated protein translocation and membrane integration, what is known about membrane protein integration, and what are main questions to address.

2) As in Point #1, a cartoon figure showing the "sliding" model described in Discussion would be helpful to improve readability.

3) Regarding Point #2. Although the authors did not explicitly mention this, the model opposed to the sliding model would be the "in-out" model, where the TMS first goes down along the translocation pore and then partitions into the lipid phase through the open lateral gate. However, in my opinion, mechanistic differences between the two models are rather small, and a boundary between the lateral gate and pore is not clear-cut (also, lipids can partially insert acyl tails into the open lateral gate and contact the polypeptide chain in the pore). The authors should clearly describe why the data fits into the "sliding" model not the "in-out" model. Also "slide along the open lateral gate" is unclear. It is possible that the TMH stays just outside the lateral gate while it slides across the membrane (similar to structures seen in studies of translocation intermediates containing TMHs or signal sequences).

4) In the previous study by Ismail et al., a bi-phasic pulling force has been observed. In the current study, the FPs do not seem to clearly show such a biphasic pattern. Can the authors discuss about this?

5) For some observations, the authors simply say further studies will be required to understand (for instance, long-range residue-residue interactions). Instead, the authors should at least provide a possible explanation or propose a testable hypothesis.

6) “Likewise, peak VI-a likely reflects the membrane integration of a hydrophobic, membrane-associated cytoplasmic segment located just upstream of TMH5, Figure 3—figure supplement 1B. In contrast, the unexpectedly high N_start_ value for peak IV indicates that integration of TMH3 commences only when its N-terminal end is ~52 residues away from the PTC possibly because of the tight spacing between TMH2 and TMH3.”. These can be perhaps tested by mutagenesis and inserting a hydrophilic stretch between TMH2 and TMH3. How could the tight spacing between TMH2 and TMH3 cause a delayed insertion of TMH3?

7) The basis for N_start_ and N_max_ for the magenta (mutant) plot in Figure 3D is not clear because there seem three local peaks. What do these local peaks and minima represent?

8) Generally, the authors only use N_start_ positions for interpretations. But interestingly, peak widths are quite variable among TMHs. Is there any correlation between the peak widths and TMH sequences (or their structural properties)?

9) Results final paragraph: It is puzzling why there is no pulling force for TMH6 but two strong peaks before and after. Can the authors offer an explanation for this observation beyond the general statement? Do mutations in the periplasmic re-entrant helix affect the peaks?

Reviewer #2:

This is an elegant report on the ubiquitous process of membrane protein insertion, studied in vivo and in vitro using the bacterial model *E. coli* – exploiting three exemplar membrane proteins of increasing topological complexity.

Our knowledge about the mechanism of membrane protein insertion is somewhat superficial and the authors are leading the way in this area. This manuscript is a prime example – and thus worthy of publication in close to its present form (certainly no more experiments required).

The exploitation of force profile analysis (FPA) developed in the von Heijne lab – now at high resolution – has proved very powerful and complementary to other techniques; such as MD simulations (also deployed here). The outcome is new knowledge about the molecular details of membrane protein insertion. Importantly the paper reveals a number of very interesting aspects of this process:

– the ability of soluble cytosolic domains to fold while still in the ribosome exit tunnel

– site specific aa substitutions can have long range effects on FPA, indicative of cooperative TMH folding and insertion.

– trans-membranes helices (TMH) very quickly lose their “pulling power”, interpreted as early membrane entry.

The results support the building evidence in favour of a “sliding model” for membrane protein insertion, wherein the incorporating TMHs are always in contact with lipids at the interface between the Sec complex and the bilayer.

I have no substantive concerns about the work. Congratulations to the authors on their study; I understand that it must have been a great deal of work.

Reviewer #3:

Technically, this is a very advanced study. However, there are technical issues with the data that are either ignored or marginalized by the authors, but that can affect the conclusions of the work on individual membrane proteins. This in particular concerns the use of Lep sequences in protein fusions to obtain more "interpretable" data.

Figure 1 displays examples for the three membrane proteins tested how FPA is applied. These are all SDS-PAGE/pulse chase measurements with membrane protein truncates arrested by a translational arrest peptide. Depending on the construct different arrest peptides are used to tune the force window. These type of experiments have been done for a very large set of protein truncates, and these data have been summarized in Figures 2-4 in position versus force graphs. The authors should make the crude data (SDS-PAGE images like in Figure 1C) accessible as an external resource so that the actual data can be inspected.

Data availability: With the current presentation, there is no way to access the quality of the data. That there are ambiguities is already evident from Figure 1C with EmrE. The arrested peptide seems shows up as two bands which signals a problem which is not further discusses. It is not clear whether the lower band is included in determining the intensity of A, nor is it clear what the nature of this protein band is. Since the crude data is not available, it is hard to access whether there are more issues with the data and how this would affect the outcome of the work.

Apparently, with GlpG, constructs with N 140-160 gave rise to multiple bands on SDS-PAGE that were difficult to interpret. This problem was solved by fusion of a portion of the Lep protein to the N-terminus of GlpG. The Lep part resides in the ribosome tunnel with the AP arrested protein constructs. This seems unsatisfactory, as the Lep sequence that is entirely irrelevant to the GlpG integration process. Since this Lep sequence apparently influences that protein band pattern, it is clearly not invariant to the overall process. Thus it is difficult to argue that with the use of the Lep extensions, meaningful data is obtained.

Relating to the above comment, also with BtuC, Lep fusions are being used. If co-translational folding in the ribosome tunnel can affect the forces measured as the authors claim, replacing parts of GlpP and BtuC with Lep will certainly affect the outcome of the work.

The authors argue that co-translational folding of the NTD of GlpG in the ribosome tunnel occurs. With folding the authors probably mean secondary structure formation? In the experimental setup, arrested chains are being used, and a long biochemical processing needs to occur before readout. How can the authors be sure that this folding is also relevant for a true, unobstructed translation/integration process? Here, the physiological relevance of the observation needs to be treated with caution.

---

## [Author Response]

Revisions:The following is excerpted from the consultation session which specifies the key points the reviewers feel need to be addressed in your revised submission. I have included the full comments of the reviewers for your consideration. However, these need not be addressed in detail in your the response accompanying the submission of your revised manuscript.There are a number of technical issues that need to be addressed in a revised text including conclusions that need to be adjusted. If co-translational folding of polypeptide segments in the ribosome tunnel can affect the integration process (which the authors claim occurs), in some cases it is problematic to use Lep fusions to solve technical issues with detection. In those cases, the native polypeptide is no longer the context for this detection issue.Perhaps the authors could show the force profiles are unaffected by the presence of the fusion proteins? Of course, if the results are affected then the conclusions should be changed accordingly.However, it may be difficult to show this convincingly given the use of Lep with Btu to allow an interpretation of the protein band pattern. The implication is that Lep affects the process of either integration or folding. Further, the authors claim sequences upstream of the N-terminus may affect integration by co-translational folding as in the case of the NTD region in GlpG. Thus the replacement of Lep for such sequences may interfere with the folding/integration process.Given the von Heijne group's previous good agreement with MD simulations, perhaps this could approach could be used to access the divergence from native behavior.Perhaps the current analysis would be acceptable for the constructs that are sufficiently long (For GlpP, N>120; a 58-residue-long LepB portion is completely exposed outside the ribosome and does not engage with SecY). GlpP is probably fine in this regard because LepB-fused constructs are N=131-224. Of course, LepB has a membrane anchor, and we don't know how that might have affected FP of GlpP. They might have some native GlpG data besides those with multiple bands, which might be useful. For BtuC, they show at least some data without LepB fusion (Figure 4—figure supplement 1B). This is nice particularly because these are shorter constructs, which could have been more affected by LepB. Nevertheless, the potential caveats in their interpretations bear correction.

We appreciate the reviewers’ concerns regarding the possible effects of the LepB fusions that we were forced (by expression issues) to make to a certain number of GlpG constructs (*N* = 131-224) and all of the BtuC constructs. However, we are confident that the LepB part does not have a strong effect on the FPs, for the following reasons:

1) For the LepB-GlpG (*N* = 131-224) constructs, the segment between Lep TMH1 and GlpG TMH1 is always 71 residues, and the C-terminal end of the LepB part is ≥70 residues away from the PTC and hence far out of the ribosome tunnel. LepB is therefore unlikely to affect the FP in the region of peak II and in longer constructs. Likewise, had the NTD been present instead of the LepB part in these constructs, its C-terminal end would also have been ≥ 70 residues away from the PTC, while its folding transition (as seen in the FP) is completed already when the C-terminal end is ~40 residues from the PTC (construct *N* = 101, Figure 3B).

2) For the BtuC constructs, the N-terminal LepB fusion part is 177 residues long, and LepB TMH1 is 155 residues away from the N terminus of the BtuC part. The ~150-residue fragment from the LepB periplasmic domain is unlikely to be able to fold, as it does not represent any compact subdomain in the protein. LepB is therefore unlikely to affect the BtuC FP in any major way.

3) Finally, we have been able to measure some points on the GlpG and BtuC FPs both for the wildtype protein and for the corresponding LepB fusions. As can be seen in the new Figure 3—figure supplement 1A (for GlpG) and the already submitted Figure 4—figure supplement 1B (for BtuC), while the *f_FL_* values are not completely identical, the shapes of the FPs are only minimally affected.

It may be helpful to include raw images and some clarification regarding statistics for the experiments where there were only 2 biological replicates..

We have included a gallery of gels from repeat experiment for all three proteins in the new Figure 1—figure supplement 1. We haven’t included all gels, as the total number of lanes would be ~1,400 (including all repeats). The statistics for the GlpG constructs for which we only have duplicate measurements is now discussed in Materials and methods. All individual *f_FL_* measurements are included in the Supplement.

The major issues summarized above may be addressed by the authors without additional experiments. Reviewer #1 offers additional experimental tests, but they are not considered essential for a satisfactory revision of this most interesting work.Reviewer #1:[…]1) The current version of the manuscript is probably difficult for scientists outside the field to follow. It would be better to extend Introduction to provide non-experts with more background information about the pathway, such as the general model for the SecYEG-mediated protein translocation and membrane integration, what is known about membrane protein integration, and what are main questions to address.

We are not very fond of wordy Introductions and Discussions, and probably err on the short side from time to time. Maybe we’ve read Strunk and White a few times too many: “Vigorous writing is concise. A sentence should contain no unnecessary words, a paragraph no unnecessary sentences, for the same reason that a drawing should have no unnecessary lines and a machine no unnecessary parts.” We’ll be happy to discuss this further, if you want.

2) As in Point #1, a cartoon figure showing the "sliding" model described in Discussion would be helpful to improve readability.3) Regarding Point #2. Although the authors did not explicitly mention this, the model opposed to the sliding model would be the "in-out" model, where the TMS first goes down along the translocation pore and then partitions into the lipid phase through the open lateral gate. However, in my opinion, mechanistic differences between the two models are rather small, and a boundary between the lateral gate and pore is not clear-cut (also, lipids can partially insert acyl tails into the open lateral gate and contact the polypeptide chain in the pore). The authors should clearly describe why the data fits into the "sliding" model not the "in-out" model. Also "slide along the open lateral gate" is unclear. It is possible that the TMH stays just outside the lateral gate while it slides across the membrane (similar to structures seen in studies of translocation intermediates containing TMHs or signal sequences).

We have considered this suggestion, but feel that putting too much emphasis on the “sliding model” would risk turning the paper into a discussion of the pros and cons of different models rather than providing a first overview of what kinds of fine-level details of the co-translational membrane integration process that FP analysis may offer. We’ll be happy to discuss this further, if you want.

4) In the previous study by Ismail et al., a bi-phasic pulling force has been observed. In the current study, the FPs do not seem to clearly show such a biphasic pattern. Can the authors discuss about this?

In the previous study, the biphasic pulling force was not resolved with the weak SecM(*Ec*) AP, and the first of the two peaks was hardly visible with the strong SecM(*Ec*-Sup1) AP. The 2-peak pattern was most clearly seen with the medium-strong SecM(*Ms*) AP (with peaks at *N_start_* ≈ 27+19=46 residues and *N_start_* ≈ 35+19=54 residues). Referring to Figure 3D, it is thus possible that peak III-b in the SecM(*Ec-Ms*) FP (green) is in fact the first peak in a biphasic peak generated by TMH2. We now mention this possibility in the manuscript.

5) For some observations, the authors simply say further studies will be required to understand (for instance, long-range residue-residue interactions). Instead, the authors should at least provide a possible explanation or propose a testable hypothesis.

At this point, our best guess is that these patterns in the EmrE FPs are explained by specific interactions between E14 in TMH1 and (polar?) residues in downstream TMHs, as stated in the manuscript. Obviously, we will try to identify such interactions in future studies. We have, however, ventured a speculative explanation for how the periplasmic surface helix located just upstream of GlpG TMH2 may affect the FP recorded for the latter.

6) “Likewise, peak VI-a likely reflects the membrane integration of a hydrophobic, membrane-associated cytoplasmic segment located just upstream of TMH5, Figure 3—figure supplement 1B. In contrast, the unexpectedly high N_start_ value for peak IV indicates that integration of TMH3 commences only when its N-terminal end is ~52 residues away from the PTC possibly because of the tight spacing between TMH2 and TMH3.”. These can be perhaps tested by mutagenesis and inserting a hydrophilic stretch between TMH2 and TMH3. How could the tight spacing between TMH2 and TMH3 cause a delayed insertion of TMH3?7) The basis for N_start_ and N_max_ for the magenta (mutant) plot in Figure 3D is not clear because there seem three local peaks. What do these local peaks and minima represent?

Good point. The discussion of GlpG peak III has been expanded.

8) Generally, the authors only use N_start_ positions for interpretations. But interestingly, peak widths are quite variable among TMHs. Is there any correlation between the peak widths and TMH sequences (or their structural properties)?

We show and discuss *N_end_* values for many of the TMHs. Similar to the *N_start_* values, the *N_end_* values tend to coincide with the C terminus of a TMH reaching ~45 residues from the PTC, hence the peak width also correlates with the length of the TMH.

9) Results final paragraph: It is puzzling why there is no pulling force for TMH6 but two strong peaks before and after. Can the authors offer an explanation for this observation beyond the general statement? Do mutations in the periplasmic re-entrant helix affect the peaks?

Indeed, this is puzzling and will require further study before anything better than mere conjecture can be proposed.

Reviewer #3:Technically, this is a very advanced study. However, there are technical issues with the data that are either ignored or marginalized by the authors, but that can affect the conclusions of the work on individual membrane proteins. This in particular concerns the use of Lep sequences in protein fusions to obtain more "interpretable" data.Figure 1 displays examples for the three membrane proteins tested how FPA is applied. These are all SDS-PAGE/pulse chase measurements with membrane protein truncates arrested by a translational arrest peptide. Depending on the construct different arrest peptides are used to tune the force window. These type of experiments have been done for a very large set of protein truncates, and these data have been summarized in Figures 2-4 in position versus force graphs. The authors should make the crude data (SDS-PAGE images like in Figure 1C) accessible as an external resource so that the actual data can be inspected.

See response to Reviewer 1.

Data availability: With the current presentation, there is no way to access the quality of the data. That there are ambiguities is already evident from Figure 1C with EmrE. The arrested peptide seems shows up as two bands which signals a problem which is not further discusses. It is not clear whether the lower band is included in determining the intensity of A, nor is it clear what the nature of this protein band is. Since the crude data is not available, it is hard to access whether there are more issues with the data and how this would affect the outcome of the work.

Additional bands running below the arrested form of the protein have been seen in previous studies and most likely correspond to nascent chains on ribosomes that are stacked up behind the one arrested on the AP. We now mention this in the legend to Figure 1. Also, we now note in the Materials and methods that *A_c_* and/or *FL_c_* controls were included in the SDS-PAGE analysis for constructs where the identities of the *A* and *FL* bands were not immediately obvious on the gel.

Apparently, with GlpG, constructs with N 140-160 gave rise to multiple bands on SDS-PAGE that were difficult to interpret. This problem was solved by fusion of a portion of the Lep protein to the N-terminus of GlpG. The Lep part resides in the ribosome tunnel with the AP arrested protein constructs. This seems unsatisfactory, as the Lep sequence that is entirely irrelevant to the GlpG integration process. Since this Lep sequence apparently influences that protein band pattern, it is clearly not invariant to the overall process. Thus it is difficult to argue that with the use of the Lep extensions, meaningful data is obtained.Relating to the above comment, also with BtuC, Lep fusions are being used. If co-translational folding in the ribosome tunnel can affect the forces measured as the authors claim, replacing parts of GlpP and BtuC with Lep will certainly affect the outcome of the work.

As explained above (and now explicitly in the manuscript), the LepB part is far outside the ribosome exit tunnel in the LepB fusion constructs, and, to the extent that we’ve been able to determine, does not have a strong impact on the FPs.

The authors argue that co-translational folding of the NTD of GlpG in the ribosome tunnel occurs. With folding the authors probably mean secondary structure formation? In the experimental setup, arrested chains are being used, and a long biochemical processing needs to occur before readout. How can the authors be sure that this folding is also relevant for a true, unobstructed translation/integration process? Here, the physiological relevance of the observation needs to be treated with caution.

This is a good point, that has been discussed in earlier studies of co-translational folding of soluble protein domains. Specifically, FPA has been shown to give results fully consistent with results obtained by real-time FRET and NMR (JMB 431:1308). Small domains have been shown to be able to fold completely inside the exit tunnel, and the NTD is small enough that it should be able to fit in the distal part of the tunnel. Moreover, as now mentioned in the text, the folding time that can be predicted for the NTD based on its “relative contact order” is ~1 msec, while the elongation cycle on the ribosome takes ~100 msec/codon. The NTD should thus be able to equilibrate between the unfolded and accessible folded states between each elongation step during ongoing translation, just as it will have time to equilibrate when the ribosome pauses on the last codon of the AP. Finally, the NTD peak in Figure 3B is seen both in vivo and in vitro, i.e, in experiments that are carried out in quite different ways.